# Mechanisms of Neutrophil Extracellular Trap Formation and Regulation in Cancers

**DOI:** 10.3390/ijms241210265

**Published:** 2023-06-17

**Authors:** Zhiyuan Zhang, Ruiying Niu, Longhao Zhao, Yufei Wang, Guangwei Liu

**Affiliations:** Key Laboratory of Cell Proliferation and Regulation Biology, Ministry of Education, College of Life Sciences, Beijing Normal University, Beijing 100875, China; 202121200029@mail.bnu.edu.cn (Z.Z.); 202221200012@mail.bnu.edu.cn (R.N.); 202221200024@mail.bnu.edu.cn (L.Z.); 202031200018@mail.bnu.edu.cn (Y.W.)

**Keywords:** neutrophil extracellular traps (NET), innate immunity, adaptive immunity, tumor, cancer, anti-tumor immunity

## Abstract

As one of the most important components of the innate immune system, neutrophils are always at the forefront of the response to diseases. The immune functions of neutrophils include phagocytosis, degranulation, production of reactive oxygen species, and the production of neutrophil extracellular traps (NETs). NETs are composed of deconcentrated chromatin DNA, histones, myeloperoxidase (MPO) and neutrophil elastase (NE), playing an important role in the resistance to some pathogenic microbial invasions. Until recent years, when NETs were found to play a critical role in cancer. NETs play bidirectional regulation both positive and negative roles in the development and progression of cancer. Targeted NETs may provide new therapeutic strategies for the treatment of cancer. However, the molecular and cellular regulatory mechanisms underlying the formation and role of NET in cancer remain unclear. This review just summarizes the recent progress in regulatory mechanisms about the formation of NETs and their role in cancers.

## 1. Introduction

The immune system is a defense network covering the whole body, with the ability to identify “self” and “non-self”, which is the key for hosts to defend against pathogen invasions and eliminate pathogens [1,2]. Neutrophils are the first group of reactive cells to play a role in defending against the invasion of pathogens (Figure 1). The immune functions of neutrophils include phagocytosis, production of reactive oxygen species (ROS), degranulation, as well as the formation and release of neutrophil extracellular traps (NETs) [3]. Neutrophils contain many microbicidal proteins, which are stored in vesicles or particles in the cell before activation. Once neutrophils sense the pathogen, these proteins are released, a process called degranulation [4]. NETs are large, sticky networks composed of deconcentrated chromatin filaments and protein molecules such as histones, neutrophil granule proteins and so on [5]. Histones play an important role in the structure of chromosomes. According to the different amino acids and molecular weight, histones can be divided into H2A, H2B, H3, H4, etc. Studies have shown that the relative levels of histone H3 and H4 affect cell growth and chromosome fidelity, and are therefore very important [6,7]. Filamentous chromatin that is surrounded by spherical proteins can help neutrophils catch and kill pathogens. Histones are the most abundant protein components in NETs. In addition, myeloperoxidase (MPO), neutrophil elastase (NE) and oncogene DEK protein (DEK) constitute the protein components in NETs structure. By inducing neutrophils with MPO-deficiency, NE-deficiency or DEK-deficiency, it is found that their NETs formation ability is inhibited. Exogenous replenishment of these substances restores NETs formation [8]. The results of a study of the antibacterial activity of synthetic NETs simulated materials against *Pseudomonas aeruginosa* show a decrease in antibacterial activity when NE was bound to the antibacterial DNA-histone complexes [9]. These results suggest that protein components play an indispensable role in the formation and functioning of NETs. NETs are playing an important role in the resistance to some pathogenic microbial invasion, and until recent years, NETs were found to play a critical role in cancer.

## 2. Discovery and Function of NETs

### 2.1. NETs

Back in 1996, researchers used phorbol 12-myristate 13-acetate (PMA) to maximally activate neutrophils and found the rupture of the cell membrane, a phenomenon markedly different from necrosis and apoptosis, and this stimulation was a new way of activating neutrophils. It was until 2004, when the formation of NETs was discovered, that the process was first named. Studies had found that when neutrophils were stimulated by interleukin-8 (IL-8), PMA or lipopolysaccharide (LPS), neutrophils were activated to form membrane protuberance and released NETs [10], and this process was called the formation of NETs, or NETosis. In 2007 [11], the structural composition of NETs was confirmed through a series of experimental findings, in which various components were also confirmed. With the progress of research, more and more protein components have been discovered, and more than 30 protein components have been found to make up NETs [12]. The role of NETs was first observed in studies of neutrophils against bacterial infections, and it remained a major component in virus [13] and fungal [14] infections. In recent years, studies have found that NETs are also involved in the occurrence and development of tumor-related diseases [15]. On the one hand, myeloid-derived suppressor cells (MDSCs) can produce NETs under the stimulation of IL-8 [16], and at the same time, in the tumor microenvironment, MDSCs can produce chemokines such as CXC chemokine receptor 1(CXCR1) and CXCR2 agonists to promote the production of NETs [17]. On the other hand, chemokines and other factors in the tumor microenvironment (TME) can recruit neutrophils to tumor areas and become tumor-associated neutrophils (TANs) [18]. TANs can also produce NETs, which play an anti-tumor or pro-tumor role in cancer disease progression. In addition, NETs are increasingly involved in autoimmune diseases such as arthritis [19], systemic lupus erythematosus (SLE) [20], and other diseases.

Only activated neutrophils produce NETs, while naïve neutrophils do not. NETs are released primarily through a cell death process called NETosis, in which the cell membrane ruptures and the cell dies. Now, of course, studies have shown that there is another way to make NETs without damaging the cell membrane, and the neutrophils retain their formal functions after this process is finished. The release of NETs and its component extracellular DNA are double-edged swords for the immune system. On the one hand, as one of the innate immune components, neutrophils release NETs in response to the invading pathogens, which helps to capture and kill pathogenic microorganisms to achieve immune protection. On the other hand, during the generation of NETs, the exposed chemical components are harmful to the hosts, because it will cause a certain degree of damage to the endothelial cells and tissues of the hosts [21]. One of the obvious functions of NETs is to help neutrophils capture pathogens. At the same time, the composition of NETs contains molecules such as antimicrobial peptides, which themselves can also play a certain role in killing pathogenic microorganisms.

### 2.2. Function of NETs

Studies have shown that NETs can help neutrophils fix and trap bacteria, fungi and viruses, thereby clearing pathogens, while mice lacking the capacity of NETs production such as deletion of genes for related proteins are more susceptible to infections, which demonstrates the important role of NETs in infectious diseases [22]. During infections, NETs can remain functional for several days until DNase I in the cytoplasm degrades them, but after the DNA components are degraded, the protein components of NETs remain until they are consumed by macrophages.

Neutrophils can kill bacteria through phagocytosis and the production of ROS, and they can use the network of NETs to capture bacteria and kill them. Different from bacteria and viruses, fungi are larger and contain hyphae or spores (the structural unit of fungi). There is no complete understanding of how neutrophils perceive microbial size, but it is clear that the release of NETs in the events of infections is dependent on microbial size. A study of human fungal infections has shown that NETs may exert antifungal effects by controlling mycelial growth, and in patients with chronic granulomatous (CGD) disease, the presence of NETs restricts mycelial growth at a moderate rate, thereby providing a degree of antifungal effect [23]. In addition, protein components such as histones, NE and MPO in NETs also play a certain role in killing and antifungal activities. In humans, it can lead to severe recurrent fungal infections if MPO is deficient, as demonstrated in mice, where MPO has been found to play an important role against large pathogens such as mycelia [24].

At the end of 2019, the Novel Coronavirus called SARS-CoV-2 pandemic swept across the world, seriously affecting our daily life and our health. Analysis of clinical data from COVID-19 patients showed that the proportion of neutrophils and lymphocytes showed an increasing trend, and the increased level of NETs, one of the markers of neutrophil activation, could also be significantly detected in patients [25,26]. What specific role do NETs play in the development of SARS-CoV-2 pandemic, whether NETs play a “main force” role in the body’s immunity and immunotherapy against the virus, and whether NETs can be used as one of the targets of disease treatment, none of this is known.

## 3. Molecular Mechanisms of NET Formation

Neutrophils can be activated in response to stimulation, and partially activated neutrophils can produce NETs. According to the production of oxidants in the process, the generation of NETs can be divided into oxidant-dependent and oxidant-independent. Similarly, according to the time sequence of neutrophil response to stimulation, it can be divided into early non-lytic and late lytic NET formation. The difference between the two is whether the neutrophil cell membrane is broken. The production of NETs in both ways is the result of activation of neutrophils, which receive infection or stimulation and recognize through various receptors, including toll-like receptors (TLR), antibody crystallizable fragment (Fc) receptors or other complement receptors, to complete activation and produce NETs [27]. It should be noted that not all NETs production leads to cell death, such as non-lytic NET formation, and neutrophils can still maintain functions and normal activities after NETs formation; at the same time, not all cell death leads to the formation of NETs, only under specific stimulus conditions. Among them, late lytic NETs formation is a slow process, because the release of NETs to the extracellular region can be detected 3–8 h after neutrophils activation, while early non-lytic NETs production is a rapid process of NETs production, because NETs production can be detected in a few minutes under the stimulation of Staphylococcus aureus [28,29]. This allows the body to fight against pathogenic microorganisms.

### 3.1. Late Lytic NET Formation (NETosis)

The classic late lytic NETs generation includes a dominant ROS dependent form and a few ROS independent forms (Figure 2). Chemical factors like LPS, PMA, and cholesterol crystal can directly stimulate neutrophils, and foreign antibodies can bind to Fc receptors on the surface of neutrophils, resulting in the release of calcium ions from the endoplasmic reticulum into the cytoplasm, and the activation of NADPH oxidase complexes through classical protein kinase C (PKC) or RAF-MEK-MAPK pathways, resulting in the production of ROS and then formation of NETs [30,31]. On the one hand, ROS can directly activate the protein arginine deaminase 4 (PAD4) to deconcentrate chromatin. PAD4, an enzyme that converts arginine to citrulline, is expressed in neutrophils and when activated, it can drive the production of NETs [32]. On the other hand, ROS can also release NE from azurophilic granules into the cytoplasm by activating MPO, where NE can bind to F-actin and degrade it to enter into nuclear [33]. In nuclear, NE can hydrolyze histones and destroy chromatin packaging, thus affecting the formation of NETs [34]. Citrullination of histone H3 is one of the markers of NETs and plays an important role in driving NETs formation. Studies showed that in mice, although NE release outside the cell was MPO-dependent, inhibition of MPO enzyme activity did not completely prevent NE release, but delayed it to some extent [35]. The effect of MPO on NETosis was reflected in the effect of MPO on NE hydrolyzed protein substrate activity. The role of NADPH-MPO-NE pathway in NETs formation has been confirmed, and the stimulation of fungi, immune complexes and other crystals can also affect NETs formation through this pathway, which has been demonstrated in mice with NADPH oxidase or MPO or NE deficiency or in patients with chronic granulomatous diseases (CGD) [35]. On the other hand, activated PAD4 can directly promote chromatin deconcentration, and then DNA is released from the nuclear, so, MPO, NE and other proteins are assembled in the cytoplasm and released from neutrophils to form NETs. In addition, activated PAD4 promotes citrullination of histones, which further promotes chromatin deconcentration [36]. The degree of the citrullination of histones demonstrates the ability of NETs production to a certain extent, and the specificity and degree of the citrullination of histones are related to inflammation caused by stimulations, which results in different levels of activation of PAD4 and production of NETs [37,38]. NE and citrullinated histones are important protein components in NETs and they are key regulators in the formation of NETs. Can they be mutually regulated? Research proves that the inhibitors used during fungal infection can block the NE signal pathway but cannot affect histone citrulline, which to a certain extent shows the histone citrulline process is not affected by NE control signal, and itself can induce NETs production. But whether the citrullination of histones affect NE-trending signaling pathway is not yet clear, so it is worth studying.

Interestingly, in some studies, when neutrophils are stimulated by ionomycin and other substances, neutrophils will respond in time to produce NETs, even though NADPH oxidase is not involved in this process, but the production of mitochondrial ROS is involved, so the production of NETs is regulated by the mitochondria in these processes. It was found that under the stimulation of ionomycin, intracellular calcium ion production would activate the calcium-activated small conductance potassium (SK) channel member SK3, which was the main component of intracellular NOX-independent pathway [37], and then activate mitochondrial ROS production. This process does not involve NADPH oxidase. Therefore, it is a NOX-independent way, but the generation of mitochondrial ROS is sufficient to induce the activation of downstream ERK and AKT, thus promoting the citrullination of histones and inducing NETosis [39]. In the process, the production of oxidant ROS also indicates that the process is oxidant-dependent, but it is also oxidant-independent in the production of lytic NETs, that is no NADPH oxidant, and no ROS production in the process.

ROS is an important indicator of NETs formation or NETosis, but how ROS induces this process is not fully understood. The chromatin untangling ability of DNA repair mechanisms was found to be a key driver of NETosis. In NOX-dependent NETosis processes, ROS first activates various MAPKs pathway cascades leading to activation of transcription factors and subsequent transcriptional excitation. At the same time, ROS oxidizes guanine. At this point, DNA damage begins to repair, and chromatin is deconcentrated [40]. In addition, the repair steps include polymerase activity and proliferating cell nuclear antigen (PCNA) interaction with DNA polymerase β/δ, which inhibits NETosis induced under certain conditions. Thus, the excess ROS produced during neutrophil activation induces NETosis by inducing extensive DNA damage and subsequent DNA repair pathways, leading to chromatin decondensed.

In 2016, it was found that the inhibitory receptor on leukocyte 1 (SIRL1) is involved in a unique signaling pathway in the induction of NETs production, that is, the process does not involve oxidant production [41]. Studies have shown that SIRL1 has a negative regulatory effect on the immune function of myeloid cells, and damage their antibacterial responses. LL-37 is a C-terminal peptide of the human antimicrobial peptide, with a size of 18KSa, also known as hCAP18 [42]. It is well known that LL-37 has antimicrobial activities against bacteria, fungi, and viruses. Similarly, there is growing evidence that LL-37 plays an important role in cancer, such as lung cancer and breast cancer, including LL-37-induced membrane receptor activation and subsequent signaling pathways leading to the change in cell function [43]. LL-37 in the human body is similar to phenol-soluble modulins (PSMs) in structure, which is a natural ligand of SIRL1. Therefore, we have reason to believe that SIRL1 may regulate neutrophils by recognizing LL-37 ligand to produce NETs, and the negative regulatory effect of SIRL1 on myeloid cells also effectively explains the inhibitory effect of SIRL1 on NETs production [44]. It was found that SIRL1 can inhibit NETs formation in specific stimulus backgrounds, including autoantibodies and specific monosodium urate (MSU) crystals. SIRL1 can significantly inhibit the formation of PMA-induced NETs but has no inhibitory effect on LPS-induced NETs because the production of LPS-induced NETs is NOX-dependent, in other words, ROS generation is induced by activation of NADPH oxidase through TLR pathway, while NETs generation induced by PMA is NOX-independent, that is, it does not pass through NADPH oxidase and does not involve ROS generation, which also explains the effects of SIRL1 on NETs is ROS independent [44]. Similarly, the study believed that SIRL1 only inhibited the production of NETs of neutrophils, but it did not affect the other functions of neutrophils. The production of NETs often causes the injury of host tissue, so the inhibition of the production of NETs protects the body from tissue injury to a certain extent [44]. At the same time, it will not affect the elimination effect of pathogen infection.

NETs production in this process occurs late and is accompanied by cell membrane rupture. As a normal physiological process in vivo, it is not clear whether this process has a certain relationship with other cell death processes in vivo, and whether the formation of lytic NETs production is regulated by other cell death mechanisms. In patients with sickle cell disease (SCD), intracellular polymerization of mutated hemoglobin can lead to the development sterile inflammation, in vivo microscopic observations of SCD mice and invitro studies of the blood of SCD patients have shown that the sterile inflammatory environment promotes the activation of neutrophil-gasdermin D (GSDMD), which triggers the release of NETs, thus resulting the development of acute lung and neutrophil-platelet aggregation [45]. We call this a GSDMD-dependent way. In addition, studies found that GSDMD-deficient mouse neutrophils can produce NETs as same as wild-type mouse neutrophils [46].

### 3.2. Early Non-Lytic NET Formation

Compared to the late lytic NETs production, the non-lytic NETs generation occurs earlier (Figure 3). Secondly, this process can maintain the integrity of neutrophils membrane, that is, it does not involve the cell death process. Another difference from the above lytic NETs production is that this process does not involve NADPH oxidase in any stimulus context and is a completely oxidant-independent type.

The similarity between them lies in that the stimulation of some pathogenic microorganisms, both of them need the relevant complement system of neutrophils and/or TLR receptors to receive the external stimulus signal, thus inducing the formation of NETs. Staphylococcus aureus can activate neutrophils via TLR2 and complement receptors, which directly activate intracellular PAD4 and stimulate the production of non-lytic rapid NETs production [29]. In addition, the non-lytic NETs production involves interactions between platelets and neutrophils mediated by TLR4 signaling, which activates platelets to recognize pathogen-associated molecular patterns (PAMPs) on the surface of pathogens and thus activates platelets [47]. Beyond this, the complement signaling pathway also activates platelets, thus initiating their interactions with neutrophils. The interaction between neutrophils and platelets depends on the presence of various molecules on their surfaces. Firstly, neutrophils express the molecule PSGL-1, which mediates the adhesion of white blood cells, and recognize P-selectin on the surface of platelets [48]. The two recognize each other and activate neutrophils to produce NETs. Moreover, the important role of mutual recognition between GPIb of platelets and CD11/CD18 antibody on neutrophils surface in the production of NETs has also been confirmed by experiments. By blocking the CD11a/b antibody of neutrophils in mice, a significant reduction in the production of NETs can be detected through confocal laser microscopy observation and statistics [49,50]. This elucidates, to some extent, a mode of interaction between platelets and neutrophils. The interaction between platelets and neutrophils is more dependent on the interaction between β-integrin and neutrophils. For example, the ability and number of NETs produced by neutrophils were significantly inhibited when treated with platelet integrin αIIbβ3 inhibitors [51]. Thus, platelets and neutrophils interact with each other and can influence the production of NETs and thus the host’s ability to combat pathogen invasion.

Previous studies have described the important regulatory role of mitochondrial ROS in the late lytic NETs production, and some studies have found that the release of mitochondrial DNA also plays an important role in the early non-lytic NETs production. Firstly, neutrophils treated with granulocyte-macrophage colony-stimulating factors (GM-CSF), a hematopoietic growth factor and then treated with TLR4 agonist showed the release of mitochondrial DNA, which is regulated by mitochondrial ROS, showing another form of non-lytic NETs formation to some extent [52,53].

How neutrophils release NETs and still maintain the integrity of the membrane has not been fully determined. Whether vesicles play a mediating role in this process, or whether the membrane spontaneously restores its integrity after the release of substances, the current study cannot fully explain. Do the way NETs are generated at different times reflect the state of the cell and its normal functions to some extent? Therefore, the study of how the cell membrane maintains its integrity during the generation of non-lytic NETs is helpful to link the late lytic NETs generation to the early non-lytic NETs production and to understand the cell states.

### 3.3. The Regulator Mechanisms of NET Formation

All physiological and biochemical processes are strictly regulated, and NETosis is no exception. Studies have shown that when the invading pathogens are small, neutrophils can clear them by phagocytosis, but when the pathogenic microorganisms are too large to phagocytosis, neutrophils play an immune clearance role by producing NETs. The key factor determining the competition between NETosis and phagocytosis is the NE, when a pathogenic microorganism is small, the phagosome fuse with azurophilic granules, which insulate NE away from the nucleus, so as to prevent the chromatin from becoming deconcentrated [54]. In this case, phagocytosis occupies an absolute advantage, NETosis is restrained, and when the pathogens individual is bigger, the process is exactly the opposite. In this sense, the NETosis process is selective.

Although the formation of NETs is strictly regulated and NETs play an important role in fighting against invading pathogens, pathogenic microorganisms can still escape the effects of NETs through a few pathways (Figure 4). First of all, the study found that the Zn2+ chelator calprotein expressed with NETs plays an important role in antifungals. It can inhibit the growth of Aspergillus at a low concentration, and it can cause fungal extreme hunger and death at high concentrations, so the study found that using appropriate calprotein inhibitors can prevent fungal induction of NETs formation, thus inhibiting the role of NETs [55]. Similarly, siacylated protein components in bacteria can directly inhibit the formation of NETs, thereby causing the escape of pathogenic microorganisms. Secondly, the study found that NETs can prevent the dissemination of pathogenic microorganisms, but bacteria and other pathogenic microorganisms will “package” themselves, wrapping themselves in a layer of capsule material for physical isolation, thus reducing the ability of NETs to act on them. Finally, microorganisms themselves can secrete endonucleases to degrade NETs. Similarly, related enzymes in microorganisms can convert the components of NETs into toxic molecules, thus affecting the phagocytosis of killer cells such as macrophages, and thus promoting their own development [56]. NETs can directly activate T cells through their protein component histones thus enhancing Th17 cell differentiation, which is mediated downstream of TLR2, resulting in the phosphorylation of STAT3 [57].

## 4. The Emerging Crucial Roles of NET in Cancers

NETs have a facilitating and inhibitory role in cancers (Figure 5). On the one hand, protein components in NETs, such as MPO, histones and proteases, can play an anti-tumor role and kill tumor cells, thereby inhibiting tumor growth and development. On the other hand, NETs themselves can also act as adhesion substrates of cancer cells, thereby promoting the spread of tumor cells, and can also promote the progression of cancer disease by promoting the metabolism of tumor cells (Figure 6) [58,59]. NETs can be found in the blood in both human and mouse cancers [60]. In mouse models of chronic myelogenous leukemia, breast cancer, and lung cancer, through isolating the peripheral blood of mice we found that neutrophils tented to release NETs more strongly than in healthy individuals [61,62]. However, in the tumor microenvironment, tumor associated neutrophils (TAN) often show that they promote tumor proliferation and metastasis through NETs formation, and the tumor microenvironment often also helps TAN NETs formation. This forms a positive feedback loop that promotes tumor growth.

### 4.1. Tumor Microenvironment Promotes NETs Formation

More and more experimental results show that the inflammatory response in the tumor microenvironment can bridge the host and cancer cells, thus affecting the metabolic cascade of cancer cells [63]. The tumor microenvironment plays an important role in tumor disease development, and NETs play an important role in tumor disease, so the link between NETs and the tumor microenvironment is particularly important.

In tumors, neutrophils have two phenotypes, namely N1 and N2. In the occurrence and development of cancer, the two phenotypes of neutrophils play completely opposite roles, N1 functions to anti-tumor, and N2 plays a pro-tumor role (Table 1). It has been shown that tumor cells overexpress an immune-suppressive cytokine—transforming growth factor β (TGF-β), which leads to neutrophil polarize into a pro-tumor phenotype, that is, the N2 phenotype. However, blocking TGF-β with the TGF-β receptor inhibitor SM16 leads to aggregation of neutrophils with an anti-tumor phenotype, so in this case, if neutrophils are removed, tumor growth is promoted [64]. These results suggest that the phenotypic polarization of neutrophils may be driven by tumor microenvironment. Whether this is related to the formation and regulation of NETs remains to be further studied. The formation of NETs has also been studied in the tumor microenvironment, and it has been found that cancer cell granulocyte-colony-stimulating factor (G-CSF) and endothelial IL-8 are the main factors promoting NETosis in tumors. The occurrence of NETosis is usually accompanied by the production of ROS. Under the stimulation of G-CSF, neutrophils accumulate in large numbers in the blood and produce ROS through NADPH oxidase, which leads to the occurrence of NETosis. Studies have shown that cancer cells can promote neutrophils to produce NETs, and by building mouse models of cancers we can find that in mouse models of related cancers such as chronic myelogenous leukemia, breast cancer, and lung cancer, neutrophils produce more NETs in cancer mice than in healthy mice, a process that is related to the effect of cancer cells on the host [61,62]. It has been confirmed that both the primary tumor and secondary tumor in mouse models of breast cancer, in their tumor microenvironment, Th2 cells can promote the metastasis of lung mesenchymal stromal cells (LMSCs) by producing IL-4 and IL-13, and the LMSCs can produce complement C3 to act on neutrophils to recruit neutrophils migration into the blood stream, causing NETs formation, which promote metastasis of tumor cells to the lung, and increased C3 levels can also be detected in sera of 41 breast cancer patients from the First Affiliated Hospital of Soochow University [65].

In the tumor microenvironment, in addition to NETs, more tumor-infiltrating immune cells play an important role, so the role of NETs and related immune cells is also extremely important in tumor progression. T cells play a major functional role in tumor-related diseases, and the role of NETs prompts us to consider a question: whether T cells and NETs play a synergistic anti-tumor role in tumors. In the tumor microenvironment, T cells are functionally depleted by chronic antigen stimulation. These depleted T cells show overexpression of inhibitory receptors and reduced production of effector cytokines, thus failing to clear tumor cells [66]. It found that T cell depletion was characterized by the detection of inhibitory receptors PD1, Tim3, and Lag3 on the cell surface. Serum and whole blood samples were taken from patients with colorectal cancer before the operation, and in the NETs-rich mouse tumor microenvironment, T cells showed a depletion and functional impairment phenotype, with increased expression of related inhibitory receptors, and depleted T cells were unable to play a role in tumor immunity [67]. Therefore, in tumor immunity, in addition to directly acting on tumor cells, NETs can also interact with immune cells infiltrating locally in the tumor to promote the proliferation and development of tumor cells by causing T cell depletion. Whether other immune cells infiltrating the tumor microenvironment have a regulatory effect on tumor related neutrophil NET formation still needs further research.

### 4.2. NET Promotes the Growth and Development of Tumor

Tumor-secreted protease cathepsin C (CTSC) also acts on neutrophils by activating neutrophil membrane binding proteinase 3 (PR3), promoting the production of NETs and thus promoting lung metastasis of breast cancer cells [68]. On the other hand, intravascular neutrophils and NETs are also the sources of tissue factors that promote angiogenesis and promote tumor cell expansion and mobility by increasing vascular permeability, the results were obtained by analyzing the colectomy tissue specimens of 10 patients with colorectal adenocarcinoma, different number of drainage lymph nodes, Caco-2 cell line and primary acute myeloid leukemia cells cultured in vitro [69]. In addition, studies in pancreatic cancer have found that the production of extracellular DNA and CXCL8 on the surface of cancer cells is increased, and the use of DNase I can degrade extracellular DNA, while the proportion of CXCL8 is down-regulated, which ultimately inhibits the metastasis of pancreatic cancer cells [70], and in this process, the transmembrane protein CCDC25 acts as the NET-DNA receptor of pancreatic cancer cells to receive signals and enhance tumor cell mobility through intracellular pathways such as ILK-β-pavin [71]. NETs immunofluorescence straining was performed on primary tumors, liver metastases, lung metastases, bone metastases and brain metastases of breast cancer patients from Sun Yat-sen Memorial Hospital, Sun Yat-sen University of China from 2007 to 2016, and it was found that NETs can promote the metastasis of tumor cells [71]. All of these suggest that NETs play a role in promoting the development of tumors.

Additionally, NETs can promote tumor cell growth by affecting mitochondrial function. As we all know, mitochondria are the energy source of the body, and tumor cells are no exception. The energy generated by mitochondria can be provided to tumor cells, thus promoting the occurrence and progression of cells. A study found that in mice with PAD4 deficiency or mice treated with DNase or NE inhibitors, through the evaluation index of the mitochondrial function and density as the mitochondrial outer membrane on the expression of mitochondrial protein transposition enzymes as well as a sign of cell proliferation, all can be found in these cases, the tumor cells are characterized by low cell proliferation, and mitochondrial function was lower than normal conditions [72]. In hepatocellular carcinoma (HCC), it could be found by isolating neutrophils in the blood of human HCC patients that NETs formation was enhanced, and NETs can trap HCC cells and then induce cell-death resistance, followed by an invasion of HCC cells and their metastatic [73].

In addition to the role played by related proteins, DNA in NETs also plays an important role. It has been found that NETs are abundant in liver metabolism and serum in many cancer patients, and NETs-DNA acts as a chemokine to attract cancer cells, which can be found as well as in mouse models. NETs-DNA activates the downstream ILK-β-parvin pathway by binding to CCDC25 transmembrane proteins on cancer cells, thereby enhancing cancer cell mobility [71].

### 4.3. NET Protects Tumor Cells

In addition to the direct promotion of the growth and development of tumor cells by NETs, NETs can also protect tumor cells by influencing the cytotoxic effects of other killer cells, such as cytotoxic lymphocytes and NK cells. CXCR1 and CXCR2 receptors are expressed on the surface of neutrophils [16], and tumor cells can secret cytokines such as CXCL1, CXCL2, and IL-8 through autocrine and paracrine, and these cytokines can bind to the corresponding receptors on the surface of neutrophils [74]. Culture of a series of supernatants from human tumor cell lines with healthy neutrophils from cancer patients showed that CXCR1 and CXCR2 agonists are the primary regulators of tumor-induced NETs production, and blocking CXCR1 and CXCR2 with corresponding inhibitors completely inhibits supernatant-induced NETs production. The regulation mechanism of NETs on tumor cells can be confirmed by cell co-culture experiments. The survival rate of tumor cells in the presence of NETs is higher than that of NK cells or CD8^+^ T cells as cytotoxic effect cells, and it can be found by using time-lapse confocal microscopy. The presence of NETs around tumor cells can significantly reduce the connection between CD8^+^ T cells and NK cells and thus play a protective role in tumor cells [17]. This demonstrates that NETs may encapsulate or shield tumor cells from the killing effect of cytotoxic effector cells. Tumor cells escape has always been a key difficulty in the treatment of tumor diseases, so the role of NETs in tumor escape therapy will become one of the targets of tumor escape therapy.

In summary, the web-like structure of NETs can enclose tumor cells, thereby protecting them from the scavenging and killing effects of other host cells such as macrophages and related cytokines.

### 4.4. Potential Intervention Strategies for Targeting NET in Anti-Cancer

The double-edged effect of NETs still holds true in cancer diagnosis and prognosis. NETs scores of 19 NETs-related genes were analyzed by LASSO Cox model, and NETs were found to be significantly associated with multiple malignant biological processes, and MPO was also associated with adverse clinical outcomes. These biomarkers represent the ability of NETs formation to predict patients’ progression [75]. IL-17 can recruit neutrophil-releasing NETs, and in addition, IL-17 blocking increases susceptibility to immune checkpoint blocking (PD-1, CTLA4) [76]. Inhibition of PAD4-dependent NETosis phenotype neutralizes IL-17, and studies have shown that the higher IL-17 and PAD4 expression in human pancreatic ductal adenocarcinoma (PDAC), the worse the prognosis, as the higher likelihood of serum NETosis in PDAC patients [77]. In cancers, NET-targeted therapy has shown success in some preclinical cancer models and may be a valuable clinical target for slowing or halting tumor progression in cancer patients. Moreover, NETs are also associated with the prognosis of cancer patients. In conclusion, NETs can be used as diagnostic and prognostic markers of cancer disease.

Current research on NETs and cancer progression is mainly focused on the positive feedback regulation between the two, that is, NETs directly affect the growth, development, and proliferation of tumor cells, while tumor cells and other immune cells in the tumor microenvironment can also directly promote the formation and regulation of tumor related neutrophil NET, which forms positive feedback promoting tumor action loop, which is very important for tumor formation and progression, so targeting NETs may become a new idea for tumor therapy. Unfortunately, in current clinical studies, no studies have shown how to target NETs to treat cancer [78]. Gonzalez-Aparicio et al. proposed a possible cancer treatment by combining immunotherapy approaches with drugs that interfere with the attraction of neutrophils and NETs release [79].

Additionally, in the process of some cancers, such as gastric cancer, pancreatic cancer and lung cancer, the formation of cancer thrombosis is the second leading cause of death in cancer patients, with a very high incidence of 15.8% in gastric cancer, 19.2% in pancreatic cancer and 13.9% in lung cancer [80,81], and NETs are also involved in cancer-related thrombosis. Currently, there are many treatments for cancers, including chemotherapy, immunotherapy and radiation therapy, etc., but many cancers have poor prognosis in the treatment process, among which drug resistance is the biggest problem [82]. Studies have shown that NETosis has an important impact on drug resistance in cancer treatment, promoting drug resistance by influencing the tumor microenvironment through various NETs component proteins.

## 5. Concluding Remarks

As we have known, in addition to phagocytosis, degranulation and ROS generation, NETs, as one of the important strategies of neutrophils against large invasive pathogenic microbes, have always been a hot spot and focus of research on the way and the effects of its action. From the initial discovery of NETs to the subsequent studies, the bactericidal effect of NETs has been widely recognized by researchers, but recent studies have found that NETs do not always play a “good” role in the process of disease, NETs have also been confirmed to accelerate the process of diseases in many cases. In different cancer models, NETs play different roles. However, in the same disease such as cancer, there are both positive and negative roles of NETs in different types of cancers. On the one hand, this is related to the microenvironment in the host, and on the other hand, it is related to the characteristics of the cancer itself. There are many substances that stimulate neutrophils to produce NETs. These stimulation conditions can cause an early or late response of neutrophils. The intracellular processes involved in the formation of NETs are different at different times. In infectious diseases, NETs generally play an important role in antimicrobial killing, helping neutrophils capture pathogens and kill pathogenic microorganisms. In cancer, the role of NETs also exists in both positive and negative aspects, and NETs themselves can also cause certain damage to host cells. Therefore, studying the specific effect and mechanisms of NETs in cancer is of great significance for the occurrence and development of cancer. The research of NETs as a therapeutic target of cancer has attracted extensive attention. In view of the double-edged role of NETs in cancer, targeting NETs will provide new strategies for cancer treatment.

## Figures and Tables

**Figure 1 ijms-24-10265-f001:**
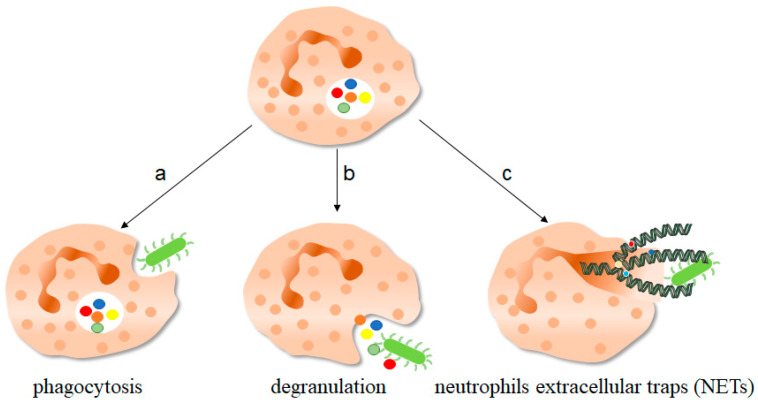
The function of neutrophils. As one of the important components of the innate immune system, neutrophils play an important role in fighting against invading pathogens. a, Neutrophils can play a phagocytic role through the depression of cell membrane, swallowing pathogenic microorganisms into the cell for treatment. b, The anti-microorganisms protein components of neutrophils stored in granules, such as defensin and cathepsin, can be released out of the cells to play a killing and scavenging role. c, Stimulated by pathogenic microorganisms, neutrophils are highly activated, which can release neutrophils extracellular traps (NETs) to capture and kill extracellular microorganisms. NETs consist of deconcentrated chromatin DNA and associated protein molecules, such as myeloperoxidase (MPO), neutrophil elastase (NE), and histones, which attach to chromatin and are released to function outside the cells. On the one hand, NETs can capture microorganisms to prevent their continued spread, thus paving the way for subsequent phagocytic killing; NETs, on the other hand, contain protein molecules that are also thought to have a direct killing effect. The green bug-like structures in the picture are bacteria. The U-shaped structures are the nucleus. The colorful spherical structures are protein molecules. The dark green spiral is the DNA molecule. DNA molecules assembled with protein molecules are NETs.

**Figure 2 ijms-24-10265-f002:**
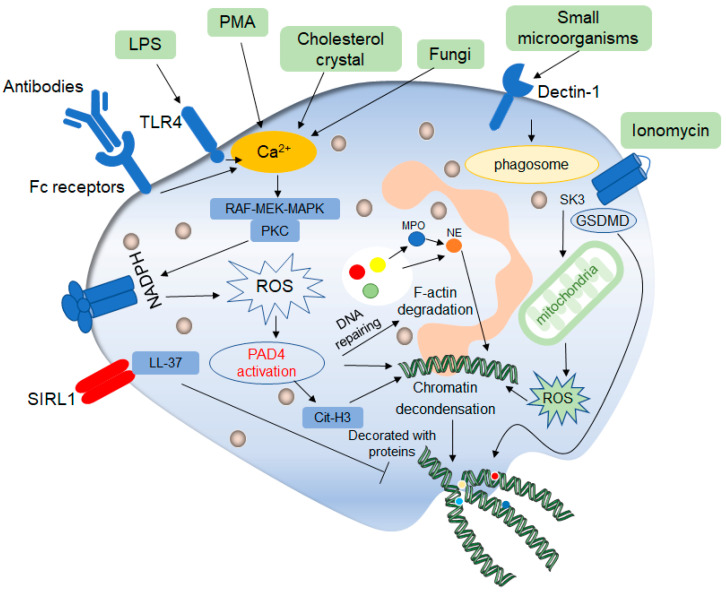
Mechanisms of classic late lytic NET formation. The classic late lytic neutrophil extracellular traps (NETs) formation is induced by PMA, LPS, antibodies, microorganisms and other substances, involving TLR receptors and complement receptors, as well as direct stimulation. Stimulated neutrophils release Ca^2+^, which activates NADPH oxidase located on the cell membrane through PKC or RAF-MEK-MAPK signaling pathways, and promote the production of ROS, thereby activating protein arginine deaminase 4 (PAD4). On the one hand, activated PAD4 directly promotes chromatin deconcentration. On the other hand, activated PAD4 promotes the release of MPO and NE, which binds to F-actin and degrades it. This process promotes NE to enter the nucleus and affects chromatin packaging. PAD4 promotes the citrullination of histones and thus affects chromatin deconcentration. Ionomycin affects the release of mitochondrial ROS through the SK3 pathway, thus affecting the packing of chromatin. The deconcentrated chromatin is decorated with protein molecules to form a network of NETs structure, after which the cell membrane breaks down, and the NETs are released to play a role outside the cell. In addition, ROS can promote the de-enrichment of chromatin by inducing DNA repair.

**Figure 3 ijms-24-10265-f003:**
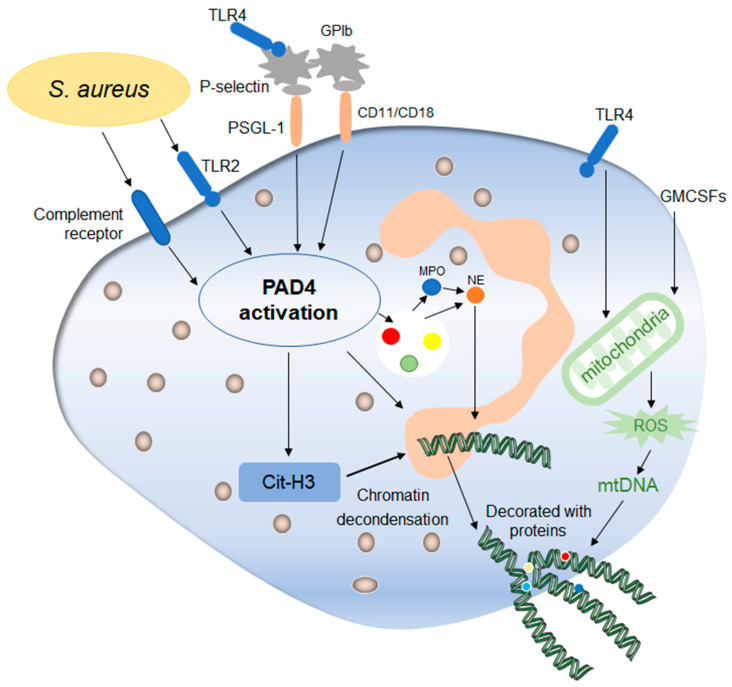
Mechanisms of early non-lytic NETs formation. Early non-lytic NETs formation is NADPH independent. Staphylococcus aureus can directly activate PAD4 by stimulating neutrophils through TLR4 and the complement pathway, which is basically the same as lytic NETs formation. In addition, platelets activated by the TLR4 signal to contact with neutrophils through their surface molecules, activating PAD4 and stimulating the production of NETs. In addition, the release of mitochondrial DNA is also one of the markers during this process, which is affected by mitochondrial ROS and then affects the release of NETs. After the release of NETs into the extracellular environment, neutrophils still maintain their structural and functional integrity.

**Figure 4 ijms-24-10265-f004:**
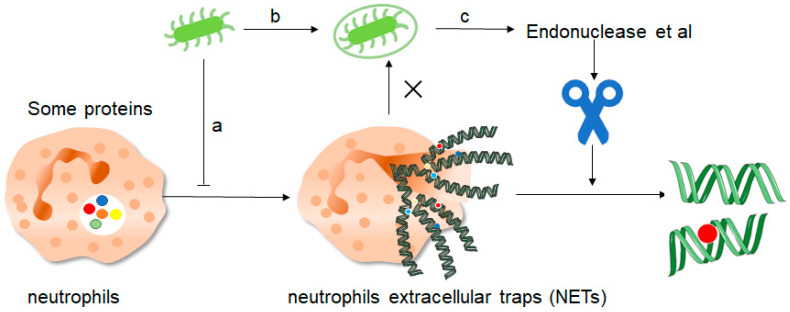
How do pathogenic microorganisms escape the effects of NETs? There are about three ways: a, microorganisms such as bacteria can produce certain components that inhibit the formation of NETs and thus promote their own development. b, bacteria can “disguise” themselves well by coating themselves with a capsule substance that shields them from neutrophil recognition and thus prevents them from capturing and clearing them. c, finally, bacteria contain some enzymes such as endonuclease, these enzymes can cut the mesh NETs into segments, which results in the loss of structural integrity of the NETs and at the same time loss of their normal function, thus preventing the role of NETs.

**Figure 5 ijms-24-10265-f005:**
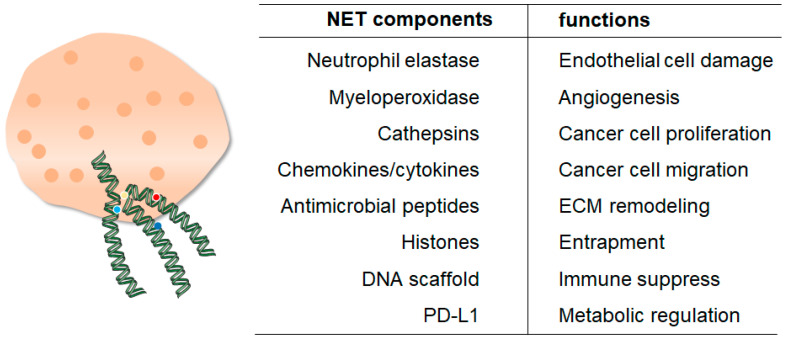
Overview of key NETs components and their general functions in cancers. The left column is the important structure of NETs, the right column is the general functions of them in cancers.

**Figure 6 ijms-24-10265-f006:**
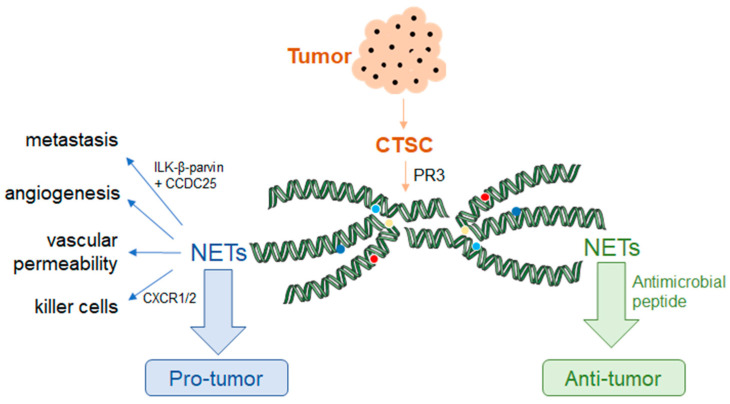
The emerging role of NETs in cancer diseases and some molecular mechanisms. The dotted line on the left indicates the pro-tumor effect of NETs, and the dotted line on the right indicates the anti-tumor effect. Tumor-secreted protease cathepsin C acts with PR3 to promote the production and release of NETs. NETs can promote the metastasis of cancer cells through the interaction of ILK-β-pavin and CCDC25 on cancer cells. In addition, NETs can promote angiogenesis and vascular permeability to play a role in pro-tumor. Moreover, NETs can influence the effects of killer cells like CD8^+^ T cells and NK cells through the CXCR1 and CXCR2 to protect cancer cells; NETs are composed of many proteins like antimicrobial peptides, which can play a role in anti-tumor.

**Table 1 ijms-24-10265-t001:** Bidirectional role of NETs in cancer progressions.

	Anti-Tumor (N1)	Pro-Tumor (N2)
**Effects** **of** **NETs** **in** **tumor**		Angiogenesis
		Vascular permeability increases
		Lymphangiogenesis
		Thrombus formation
	Tumor capture	Reconstruction of extracellular matrix
	Limitation of tumor growth	Production of pro-tumorigenic cytokines
	Proteins of NETs—anti-tumor	Promotion of tumor cells metastasis Promotion metabolism of tumor
	Inhibition metastasis of cancer cells	Promotion
		Shield tumor cells from killer cells
		Tumor microenvironment promote NETs production
		Affecting mitochondrial function

## Data Availability

Not applicable.

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
