# Peer review of "Mechanisms of Neutrophil Extracellular Trap Formation and Regulation in Cancers"

_ijms, 2023, doi:10.3390/ijms241210265_

Round 1

Reviewer 1 Report

The presented paper by Zhang and colleagues discusses the role of NETs in the disease, with the main focus on cancer. Since the discussed topic is a pretty emerging field of medicine nowadays, I went through the draft with great interest. 

I found the paper well structured, fairly written in terms of flow and readability as well as comprehensively discussing the (pato)biology of NETs. I am leaning towards acceptance, albeit, some major changes must be done.

Firstly, I miss some brand new evidence of molecular basics for NETs generation.  Recently, some new mechanisms came into play: GSDMD-dependent NETs generation under proinflammatory stimuli (https://doi.org/10.1182/blood.2021014552) that is very relevant to cancer; usage of synthetic NETs mimetic materials (DOI: 10.1126/sciadv.adf2445) or influence of NETs on Th cells through TLRs modulation (DOI: 10.1038/s41467-022-28172-4) These papers should be broadly discussed. 

Moving further (even for narrative reviews), there is a very limited amount of information given regarding the following PRISMA guidelines and description of the searching strategy, which is so crucial for review papers. The Author should include at least: Data sources and searches, Study eligibility criteria, Study selection process, Data extraction, and study quality assessment (assessing the risk of bias (ROB) for each included study), Data synthesis. MeSH terms (in addition/replacement of keywords) are necessary to be included. For each step, it is necessary to explain to the reader with pictures or tables. It is necessary to explain what was drawn at each step to lead to the result. Moreover, a figure showing the PRISMA-based workflow must be drawn accordingly to the Prisma schema. After that, a discussion is valuable even for narrative papers. A description of the Data Mining strategy should also be included.

Since Histones play a vital role in NETs generation/biology I would like to suggest introducing more information about them - their classification, the difference between e.g. H2A and H4 and so on.

I would also suggest creating the figure which will bring some new mechanistic approaches to the process of NETosis - the presence of a good figure sometimes says more than 1000 words. It will also increase the visibility and citation of this work. 

There are some minor grammatical errors / minor typos - please go over them carefully. (e.g. line 476)

There are some minor grammatical errors / minor typos - please go over them carefully. (e.g. line 476)

Author Response

Please see the attachment。

Reviewer 2 Report

In this article, Zhang et al. review the role of neutrophil extracellular traps (NETs) in cancer development and progression. NETs are networks composed of deconcentrated chromatin filaments and protein molecules that play a critical role in the resistance to pathogenic microbial invasion. The tumor microenvironment promotes NET formation, and NETs can promote tumor cell growth and development. The review summarizes the recent progress on the regulatory mechanisms of NET formation and its role in cancers. NETs have both positive and negative roles in cancer development, and targeting NETs may provide new therapeutic strategies for cancer treatment. The authors needs to address following concerns for the final acceptance.

Major concerns:

  1. The review primarily focuses on the general mechanisms of NETs. However, the role of NETs in specific cancer types is essential, which will help highlight any differences in their function or regulation.
  • The review needs additional sections to discuss the clinical implications of NETs in cancer diagnosis, prognosis, or therapy.
  • The review also needs precise molecular and cellular mechanisms underlying the formation and role of NETs in cancer, which could help in the identification of new therapeutic targets.

Minor grammatical errors including punctuation errors, spellings, and spaces. 

Round 2

Reviewer 1 Report

The Authors correctly addressed all my majors. No further comments. 

Reviewer 2 Report

The authors have responded to the concerns satisfactorily.